# Failure of a Multi-Centric Clinical Trial Investigating Neoadjuvant Radio-Chemotherapy in Resectable Pancreatic Carcinoma (NEOPA-NCT01900327)—Which Lessons Are Learnt?

**DOI:** 10.3390/cancers15174262

**Published:** 2023-08-25

**Authors:** Michael Tachezy, Florian Gebauer, Emre Yekebas, Jakob Robert Izbicki

**Affiliations:** 1Department of General, Visceral and Thoracic Surgery, University-Hospital Hamburg-Eppendorf, Martinistrasse 52, 20246 Hamburg, Germany; florian.gebauer@helios-gesundheit.de (F.G.); eyekebas@yahoo.com (E.Y.); izbicki@uke.de (J.R.I.); 2Department of General and Visceral Surgery, HELIOS University Hospital Wuppertal, 42283 Wuppertal, Germany

**Keywords:** prospective randomized trial, pancreatic cancer, neoadjuvant therapy, radio-chemotherapy

## Abstract

**Simple Summary:**

The NEOPA trial was a large clinical trial in Germany, funded by the German government. It aimed to investigate if a treatment before the surgical removal of pancreatic head cancer, consisting of a combined local radiation and chemotherapy, might improve the survival of the patients. The trial had to be discontinued due to recruitment failure. In this article, the reasons for this are investigated and discussed, intending to draw lessens for future research initiatives and to address deficiencies in the surgical study culture.

**Abstract:**

Background: A significant number of clinical trials must be prematurely discontinued due to recruitment failure, and only a small fraction publish results and a failure analysis. Based on our experience on conducting the NEOPA trial on neoadjuvant radiochemotherapy for resectable and borderline resectable pancreatic carcinoma (NCT01900327—funded by the German Federal Ministry of Education and Research—BMBF), we performed an analysis of potential reasons for recruitment failure and general problems in conducting clinical trials in Germany. Methods: Systematic analysis of environmental factors, trial history, conducting and funding in the background of the published literature. Results: The recruitment failure was based on various study-specific conceptional and local environmental aspects and in peculiarities of the German surgical study culture. General reservations against a neo-adjuvant study concept combined with game changing scientific progresses during the long-lasting planning and funding phase have led to a reduced interest in the trial design and recruitment. Conclusions: Trial planning and conducting should be focused, professionalized and financed on a national basis. Individual interests must be subordinated to reach the goal to perform more relevant and successful clinical trials in Germany. Bureaucratic processes must be further fastened between a trial idea and the start of a study.

## 1. Introduction

The NEOPA trial was a study funded by the German Federal Ministry of Education and Research (BMBF) on neoadjuvant radio-chemotherapy for resectable and borderline resectable pancreatic carcinoma, which was initiated at the beginning of 2014 and had to be prematurely stopped after nearly three years and recruiting 30 patients at the end of 2016 due to insufficient recruitment [1]. In total, 410 patients were planned to be recruited in 16 study sites. In an international analysis, between 25% and 50% of the clinical trials are terminated before the study goal is reached due to a lack of recruitment [2]. Another large proportion of the completed trials require a longer recruitment phase than planned, so the termination of the NEOPA trial was not a particularly unusual occurrence [2]. Some time has passed since then, but in our opinion, it is still necessary to reflect on the circumstances and possible reasons for the failure of the study in order to be able to draw lessons for the planning and implementation of future projects but also to leave an impact for thoughts or impulses for possible general or systemic obstacles in planning and performing a large clinical trial [3,4]. 

## 2. Background and Study Development

When the planning for the NEOPA study began in 2009, there were only a few retrospective studies on the concept of neoadjuvant therapy in pancreatic cancer. Almost all of them came from the United States, where neoadjuvant therapy was more widely accepted as compared to Europe [1,5]. Established non-surgical, additive therapies were still in the ‘adjuvant gemcitabine mono phase’—that lasted for so many years [6]. Thus, the idea of neoadjuvant radio-chemotherapy, at least in Germany, for resectable and borderline resectable pancreatic head carcinomas was still relatively bold—but not unique! Before the NEOPA trial was planned, two European initiatives, one mono-centric study in Italy and another multi-centric trial in Germany, have already failed to recruit in similarly concepted studies and basically struggled with similar problems [7,8]. Notwithstanding, the Hamburg trial team assessed the situation at the time, which was that the trials idea was still highly relevant, and the time had come for such a neo-adjuvant concept. During exploratory interviews, the willingness to participate in the project of the largest pancreas centres in Germany was evaluated, and there was, in principle, a positive opinion about the realization and participation in the study. We thought that we might have a better position in the oncological and surgical society to motivate the potential co-workers and successfully conduct and recruit the study. 

The experience of the Hamburg study team consisted of participation at uni- and multi-centric trials (e.g., ESPAC etc.) but minor experience in the planning and implementation of a multi-centric investigator-initiated trial (IIT) as a leading study centre. The leading investigator, Professor Emre Yekebas and his team consisting of Florian Gebauer and Michael Tachezy, submitted the first study outline to the Federal Ministry of Education and Research (Bundesministerium für Bildung und Forschung, BMBF) in the fall of 2009. During all phases of the study, the clinical team was supported and advised by a clinical research organization (CRO) associated at the university hospital—a CRO with a lot of experience in phase I and II trials, but less experienced in larger principal investigator-initiated, non-industrial phase III trials. During the planning phase, the team was advised by a consortium of internal and external experts in the treatment of pancreatic cancer and conducting oncologic trials. 

As a result of the review process, a re-submission was recommended in the call in 2010 after requested changes had been made. After a new evaluation, an invitation was made to submit a full application, which was approved in end of 2011. Another formal application was necessary, and ultimately, the funding commitment was made in June 2013 (Figure 1). 

Another year and a half passed between the approval and the start of the study, since according to the specifications of the sponsor, an application to the ethics committee and the Federal Institute for Drugs and Medical Devices (Bundesinstitut für Arzneimittel und Medizinprodukte, BfArM) before the funding decision was finally made was not permitted. University Hospital Hamburg-Eppendorf was initiated as the first centre in February 2014. By this time, Professor Yekebas had already left Hamburg, and Professor Izbicki—as head of the clinic—took over the role of principal investigator. Professor Yekebas—now head of surgery of the clinic Darmstadt—participated with his new clinic as a study site and remained the “scientific study director”. In the following years, another 15 centres were initiated, but despite the efforts of the study management, recruitment did not pick up speed, and even an extension of the inclusion criteria as part of a substantial amendment (March 2016) could not help to improve the study, so it had to be cancelled in November 2016. 

At the time of the trials planning, no routine checklists for assessing clinical trial feasibility existed to our knowledge or were used to critically reflect the recruitment potential [9]. It was recently postulated that this should be mandatorily done during a funding application process [10]. Besides a systematic approach, several advisors in the oncological, surgical radio-oncological community were contacted. However, the scientific and educational work on trial planning and management significantly increased was during the recent years and was discussed on a wider base [11].

Another extraordinary aspect is the fact that the ‘PREOPANC’ study (EudraCT: 2012-003181-40) had an almost identical study protocol and was successfully carried out at around the same time (2013–2017) at 16 centres in the Netherlands and published in 2020 [12]. This might indicate that specific aspects in Germany, the German surgical university and hospital landscape and the health care system (for example, the culture of funding, research and centralization) might have a considerable share in the success or failure of a clinical study, and a comparison to the Dutch and similar systems is undertaken. Another interesting fact, highlighting the multifactorial genesis of the recruitment failure, is that even the recruitment of the Hamburg study team had significantly stayed behind the expectations (Figure 2). 

Interestingly, similar problems emerged years later in the NEONAX trial, which investigated the impact of neoadjuvant chemotherapy on overall survival. Although the trial was eventually successfully completed, it also experienced major recruitment difficulties which resulted in a reduction in the number of patients and an adjustment of the statistical calculation of the primary endpoint [13].

Various publications investigated and described possible reasons for recruitment problems [14]. In a systematic analysis of RCTs that were discontinued due to the recruitment, the reasons were grouped in four different aspects: funding, design, recruiter or participant related [3]. An interview-based study of clinical trial stakeholders added to the forementioned aspects of the research environment, design and trial team related factors [3]. 

We conducted an analysis of the processes during the phases of the study, the development of the study hypothesis, the application process and study initiation, the execution, adjustments and premature determination. Moreover, environmental factors at the local level, but also in general in the study—and funding—system in Germany, are investigated and discussed against the background of the published literature. The analyses include all study sites: some did not recruit at all, but the recruiting centres were significantly behind the expectations—even our own institution.

### 2.1. Study Design

Overall, compared to similar studies such as the ESPAC trials, the NEOPA trial was not an exuberant study with regard to the inclusion and exclusion criteria, but a lot of items were planned to be documented. An interviewee in the publication of Briel and colleagues described this phenomenon well: ‘The study protocol is written by scientist not enough aware of the pitfalls conducting a trial and too much details are included in the protocol’ [14]. Additionally, the peculiarities of the study following the German Drug law made a great deal of documentation and an amount of personal time necessary. 

However, contrary to the majority of trials, the overestimation of the prevalence of patients was not the major problem, but the inclusion and screening process [3] were. One of these problems was the need for a pre-therapeutic histological confirmation (fine needle aspiration via endoscopic ultrasound—FNA EUS), which was not generally established and was a deviation to the clinical routine [7]. It is therefore a relevant deviation from the clinical standards, and this reduces the motives for engagement [15]. From a medical point of view, and with this reducing the acceptance of researchers, it has weaknesses in diagnostic accuracy. And at least at that time, it was not considered undisputed, since a tumour spreading, acute pancreatitis and bleeding with the subsequent negatively affected operative site were feared [16]. 

Another factor that highly influences the degree of patients’ willingness to participate in a study was the concept of non-surgical pre-treatment and randomization in general [17]. Other studies have shown that the patient’s impulse to want to have a tumour removed as soon as possible leads to significantly poorer acceptance and corresponding recruitment in a study, which can be influenced by good information and persuasion but is part of the concept and not conceptually fixable [8]. 

### 2.2. Environment

#### 2.2.1. Duration between Idea and Start of the Trial

With good reason, the organization of clinical studies is linked to high regulatory requirements, which, however, increasingly result in bureaucratic requirements that are not always effective [18]. These bureaucratic obstacles in applying for and conducting a multi-centric study, coupled with relatively low experience as the lead study centre, led to a significant delay in the progress of the project.

The two-stage application took approximately 2 years from submission to the commitment of funding and, after that, another year to the actual funds release (Figure 1). Even when deducting the 12 months that resulted from the resubmission, the approval process seemed too long for us to be able to answer relevant clinical questions in a timely manner [14].

At the time, the regulations of the sponsor involved that applications to the lead ethics committee and the Federal Institute for Drugs and Medical Devices (Bfarm) could be submitted after the final funding decision had been received. This application process alone took 21 months due to the complexity of the study and multiple inquiries. Facing these regulatory issues, the European Clinical Trials Regulation, adopted in 2014, was introduced this year and aimed to centralize and harmonize the submission, approval and monitoring processes, shorten the deadlines and reduce bureaucratic costs [19]. However, its functionality in practice must be proven in the future. An encouraging example would be the amendment of the German radiation protection aid (in 2019) that has significantly increased the approval time [18]. 

Another underestimated time factor was the processes for approval and initiation of the centres: obtaining the necessary documents from the respective investigators and trial sites for submission to the local ethics committees, the process to proof the contracts and signing processes usually took longer than expected [18]. The local checks of the vote of the central ethics committee and the suitability of the local study centres consumed a lot of time, so that the study centre initiations, for which an appointment for a visit had to be found and planned, dragged on for months (see Figure 1). The basic reasons were that recurring central documents, such as the investigator’s curriculum vitae or the study centre description, were not submitted in a timely manner. Moreover, the documents and signatures are still provided on paper and a switch to digital signatures and documents might lead to a reduction in turnaround times [18]. A positive example is the new policy of the National Cancer Institute that requires a single institutional review board of records be used for all protocols for multi-site research, which has significantly sped up the approval processes in the United States [20]. 

The regulations and guidelines to conduct a clinical trial are, at least in Germany, so complex and extensive, that even for professional CROs and industry driven projects, the planning and conducting of a clinical trial are challenging, and semi-professional teams such as university-driven projects and IITs might possibly be overstrained [18]. 

As shown above, more than five years passed between the conception and the initiation of the study. At the time of planning and initiation, combined neoadjuvant radio-chemotherapy was a phenomenon that was largely carried out in the USA; the European and German centres and guidelines recommended immediate exploration and resection [1,5]. The acceptance of a neoadjuvant concept for resectable and borderline-resectable pancreatic carcinoma was low in Germany at this time. 

In this period, the pioneering publications by Conroy and colleagues and by Hoff and colleagues in the palliative setting on FOLFIRINOX and gemcitabine/paclitaxel were made [21,22]. The idea of the study protocol with a radio-chemotherapy was no longer en vogue at the time of initiation, and due to the groundbreaking study results, chemotherapy alone was preferred in the experimental neoadjuvant setting [23]. 

To face the various problems of an RCT such as the NEOPA trial, pilot studies applying the planned informed consent procedure might be a possible solution to avoid recruitment failures, but it must be mentioned that it further delays the start of the main trial [3]. 

#### 2.2.2. Study Culture in Surgery in Germany

There are 35 university clinics in Germany with at least as many surgical clinics and far more clinics with comparable capacities. Much has been written about the problem of the lack of centralization in Germany [24]. From our point of view (and this is based on the current evidence), this is an undoubtful factor in the quality of patient care and clearly leads to a poorer research culture compared to the more centralized countries of Europe, such as the Netherlands or Scandinavia [14]. Few specialized centres treat basically all patients with a specific disease and can plan and conduct clinical trials on them. The increase in the minimum number of cases per centre could result in an increase in the number of study patients in addition to an increase in quality of treatment. 

Taking part in studies for certification purposes only seems to have a minor impact, especially since less complex registry studies and smaller and less workload bearing in-house studies can be used instead. 

The scientific value of a study is not the only motivator for or against active recruitment. Economic interests, as well as numerical prestige, play a relevant role in these considerations. Case or surgery-related payments for the hospitals as part of a DRG system, in the case of private health insurance even directly to the chief physicians, might have an impact on the decision for or against participating in a study if there is a chance that a patient might not to be operated on (e.g., due to progress in the context of a previous therapy). Studies with surgical questions that do not change the basic therapy (two neoadjuvant regimens, e.g., ESOPEC) or surgical techniques recruit well and quickly (e.g., DISPACT, PROUD) [25,26].

A relevant factor is also the personal commitment of the PI and his network. Each centre active in clinical studies has its network, that is mostly grown and based on personal relationships, which are influenced to a greater or lesser extent by particular interests [14]. In Germany with its 82 million citizens and 19,000 newly diagnosed cases of pancreatic cancer per year (of them approximately 50% with stage I–III), around 50 major surgical centres (with 50 and more pancreatic resections per year) compete to treat patients with pancreatic carcinoma [27,28]. Usually, several studies that are recruiting for these patients are active at the same time, which means that they are in relevant competition with one another. Here, a resilient network or a profound conviction of the participating centre investigators must exist in order to have the study recruited by the other centres. Centres too often take part out of a supposed “favor” or political motivation and recruit below prognosed numbers, so that in the end, there is a lot of effort and costs with little profit (=recruitment). The recruitment estimates collected prior to study participation and the start of the study were blatantly missed, so in the NEOPA study, only 7 out of 17 active centres included patients. 

Just recently, namely in 2017, the association of surgical oncology (ACO) was founded with the aim to form a society to ‘bundle the activities of the visceral-surgical oncology and to optimize the cooperation between the Society of General and Visceral Surgery (DGAV) and German Cancer Society (DKG)’. With its working groups, surgically relevant trials shall be identified and created [29]. However, whether these goals can be reached by overcoming the individual interests and creating a collective motivation has not yet been shown, and further efforts must be made to achieve a rethinking of the influential players in the surgical society. The competition already described might only continue here—people rarely pull together.

Taking the Netherlands as an example, multi-disciplinary study groups such as the Dutch Pancreatic Cancer Group (DPCG) were established in 2011 [30]. The goals are an export of medical scientific research, a quality improvement through by a central organized registration and auditing to promote scientific education, organize conferences and publish publications on pancreatic cancer and to provide and cooperate with training courses. During only 10 years, an impressive number of trials were and are currently conducted [29,31]. They have realized the creation of a central working group in which all local investigators work together from the start and are involved in the development of the protocols, and accordingly, the participation in the hospitals is high. They have defined clear rules regarding authorship. 

Similar initiatives to intensify the cooperation between groups were started in the last ten to twenty years in different countries, such as the United Kingdom Surgical Trials Initiative that covers nearly all General Surgery Units in the UK [32]. Other examples of nation-wide cooperation networks are the Clinical Trials Network Australia and New Zealand. In the US, no comparable organization on a national basis exists. This might be caused by the heterogenous system of highly variable and non-standardized hospitals that are either public or profit orientated who maintain their own records and referrals. As such, there is no easy way to combine data sources nationally or from a government level, except the Veterans Affairs hospital network, where only a small proportion of highly specialized complex care occurs. The situation in the US might be similar to the German dilemma of high competition between the institutions. Consequently, trial organization and execution are reliant on multiple small academic individuals and groups getting together, bringing their own researchers and fellows. The system is much more discontinuous, but data can still be generated with large numbers. 

Other interesting networks are based on medical student and trainee-led surgical research networks such as the EuroSurg or STARSurg network. An US network is the West Midlands Research Collaborative (WMRC) that is a trainee-led collaborative. The latter underlines the importance to integrate and train young colleagues from the beginning of their career and to view surgical research as a natural part of the work [32]. 

Another drawback—at least in Germany—prohibiting the ideas of a national study society that conducts clinical trials is that the funding criteria allow a funding only at public health facilities, and an independent organization is exempt from this [33]. Initiatives such as the National Surgical Trial Network (CHIR-Net), which have been trying for many years with great commitment and enthusiasm to promote cooperation between the centres and to increase the study quality and funding application approval rate, have not yet been able to exploit their outstanding potential due to a lack of financial, personal and organizational support [34,35,36]. 

#### 2.2.3. Trial Team and Staff Motivation

At the beginning of the NEOPA trial, a well experienced study nurse completed the study team and was funded over the complete time during the trial. During the first year, a medical study coordinator was full time part of the study group, but in the following years, the study coordination was done beside the clinical work. The scarce experience of the medical core team in conducting a multi-centric clinical trial was already described. The increasing complexity of clinical studies requires co-workers who are exclusively involved in conducting studies and are appropriately trained [3,37]. Despite the fact that the current exodus of nursing staff from the bedside to administrative jobs is improving the situation on the job market for study staff, many clinics do not have any permanent positions for study staff and have to finance their study nurses from third-party funds (which accordingly significantly reduces the attractiveness of the often timely limited positions). The patient case payment of the IITs, such as the NEOPA trial, only represents an expense allowance, which compensates basically the costs incurred but certainly does not represent a monetary motivation. Of course, the motivator for recruitment in a study should not be money (and most of the scientists pretend that it would be the case), but in times of limited financial resources and lacking posts in the hospitals, this might be an important driver [15]. 

Not only is the design and application of clinical studies extremely laborious, but so is their implementation as a participating centre. So how is the motivation of the medical staff to be guaranteed? The majority of scientists proclaim that the gain of knowledge and the improvement of healthcare and quality of methods is the main driver for study participation and should be enough motivation; however, another and nearly as important factor is the personal benefit [15]. When such large studies are published, usually only the investigators and trial deputies responsible in the front row (director: in the clinic/senior physicians) are co-authors on the publications, with the main work getting stuck with the subordinate assistants [14]. Even if the latter are taken into account, the publication outcome in the surgical journals is usually lower than that of basic research and associated with a longer time horizon [38]. Fewer and fewer employees of the following generations are willing to pay in advance in order to hope for the promised later merits/funding of the immediate beneficiaries. Scientific commitment should, but does not, necessitate a more intense clinical training and support in all clinics and vice versa. Another circumstance, which will not be discussed in detail here, is the loss of attractiveness of top clinical positions in Germany, not only in non-public hospitals but also in universities, that leads to a lower rate of scientific work as entry in these top positions. 

Large oncological studies are usually not only complex in terms of treatment but also in all in terms of documentation and follow-up, so that even well-crafted and lean studies can have difficulties to recruit. In addition to the (already described) limited motivation of the study staff, there is also the ambition to fulfil the requirements of a trial in the best possible way, which further limits the capacity to recruit within a certain time frame (plus learning curve of the study staff, etc.).

Without wanting to go into the subtleties of study planning, the situation can arise that studies based on the existing (weak retrospective) data are designed as superiority studies, although the planners of the study do not consider the regimen to be more effective. This might be done for pragmatic but statistically incorrect reasons for calculating the number of cases (a non-inferiority study usually requires far more patients) with the result that even the PI of a study does not believe in the basic scientific assumption of the study, which can influence his commitment to the study. Another problem that might arise during the study period is the leaving of the PI: the funding and, thus, the study management are tied to the institution but can be transferred to another institution if the applicant and PI change. However, since the financial overhead flows to the institutions, there might be an interest in not letting the study go.

#### 2.2.4. Funding 

In contrast to a lot of other promising trials, the funding per se was not a problem of the trial. The study was always well financed, not as good as compared to industry-funded trials, but still good compared to other IITs. 

However, there might be a general problem or shortage of trial funding in Germany. Apart from the major sponsors DFG/BMBF and German Cancer Aid, there are hardly any larger foundations that could fund multi-centric study projects. The institutions mentioned almost exclusively provide support programs for clinical trials without a specific subject, so that there is always an enormous competition with the other medical disciplines. In recent years, the general and visceral surgical societies have failed to establish themselves in the review system, so that there does not seem to be sufficient support or advocates in the foundations. The lack of consensus in the surgical committees also leads to a lack of agreement and the dispatch of surgeons to the funding institutions.

Another possible point could also be the reviewer system. In principle, national surgical experts are possibly not unbiased, and the international experts probably have a good overview of the German study and health system, which could lead to potential misjudgements of the local needs.

## 3. Conclusions

The reasons for the NEOPA trials failure are numerous, but one of them was that the short time frame of scientific interest on the topic was missed due to long processes until the start of the trial and coincidental scientific progress. 

In general, conducting surgical clinical trials requires an enormous personal expertise and motivation and needs to be more appreciated and professionalized at least in most of the German surgical departments. Institutional and individual barriers need to be taken down to take full advantage of the enormous potential of the German surgical community. There is a need of a nation-wide and interdisciplinary coordination of clinical studies in specific diseases and indications—for example, controlled by the surgical working groups in cooperation with an appreciated national study network CHIR-Net. In this way, the particular interests of the respective big players could be reduced. During the last years, regulatory improvements were initiated in Europe and will hopefully optimize and speed up the processes. 

## Figures and Tables

**Figure 1 cancers-15-04262-f001:**
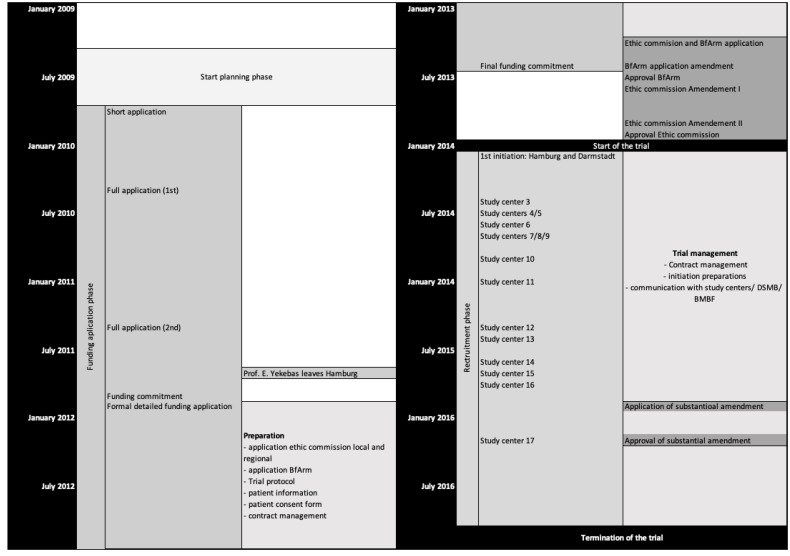
Study timeline with milestones.

**Figure 2 cancers-15-04262-f002:**
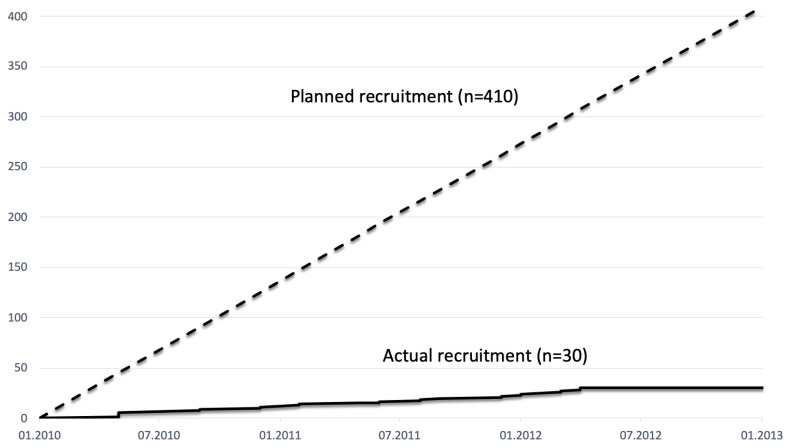
Actual and planned recruitment curve.

## Data Availability

The datasets used and analysed during the current study are available from the corresponding author on request.

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
