# Peer review of "Failure of a Multi-Centric Clinical Trial Investigating Neoadjuvant Radio-Chemotherapy in Resectable Pancreatic Carcinoma (NEOPA-NCT01900327)—Which Lessons Are Learnt?"

_cancers, 2023, doi:10.3390/cancers15174262_

Round 1
Reviewer 1 Report
Very good and important study. Congratulations!
Author Response
Dear Reviewer, thank you very much! We are happy when we get the chance to share our experience and analysis!
Reviewer 2 Report
The paper refers to the causes of the failure of the NEOPA trial, reflecting on the situation in Germany and the research and development process. As a fellow researcher, I found it very interesting.
1) Although the data only shows the number of recruits, if the reasons for this are given by the institutions that did not recruit any patients, this should be stated.
2) The descriptions are separated by content, but there is some crossover between content, which makes it difficult to understand what the author is trying to say, so the descriptions could be made more clear.
In particular, (A) and (B) under 'Enviroment' both seem to describe the specific German environment.
3) On page 7, a comparison is made with the Netherlands, but a comparison with the USA and other Asian countries would also give a clearer characterisation of Germany.
4) The rightmost number (02.2016) on the X-axis in Figure. 2 is difficult to read.
Author Response
1) Although the data only shows the number of recruits, if the reasons for this are given by the institutions that did not recruit any patients, this should be stated.
Ad 1) This is a good point and should be mentioned: Some trial sites did not recruit at all but all the others were significantly behind the expectations – even our own institution. So, the described problems and reasons for this include all sites. We the following in the introduction:
‘The analyses include all study sites: some did not recruit at all but the recruiting centers were significantly behind the expectations – even our own institution.’
2) The descriptions are separated by content, but there is some crossover between content, which makes it difficult to understand what the author is trying to say, so the descriptions could be made more clear.
In particular, (A) and (B) under 'Enviroment' both seem to describe the specific German environment.
Ad 2) You are right, both A and B describe the situation in Germany. However, we wanted to address two different problems: The reasons for the delay or prolongation of the processes until the start of the trial, that are mainly caused by bureaucracy and the long process of funding decision (A) and (B) the specific situation of surgical research in Germany headlines as ‘culture’, the cooperation deficits due to competition between the universities and the lack of motivation to participate and recruit actively in multicentric trials.
3) On page 7, a comparison is made with the Netherlands, but a comparison with the USA and other Asian countries would also give a clearer characterisation of Germany.
Ad 3) Thank you for your helpful comment, we added some information on network projects and the situation in the US.
4) The rightmost number (02.2016) on the X-axis in Figure. 2 is difficult to read.
Ad 4) Changed accordingly
Reviewer 3 Report
Very interesting paper on an under discussed topic. Reasons for the failure of the trial are extensively explored with great honesty. The article should be read by all young investigators before launching a trial!
Author Response

(The authors gave the same response as above.)
